# The Utilization and Benefits of Telehealth Services by Health Care Professionals Managing Breast Cancer Patients during the COVID-19 Pandemic

**DOI:** 10.3390/healthcare9101401

**Published:** 2021-10-19

**Authors:** Donovan A. McGrowder, Fabian G. Miller, Kurt Vaz, Melisa Anderson Cross, Lennox Anderson-Jackson, Sophia Bryan, Lyndon Latore, Rory Thompson, Dwight Lowe, Shelly R. McFarlane, Lowell Dilworth

**Affiliations:** 1Department of Pathology, Faculty of Medical Sciences, The University of the West Indies, Kingston 7, Jamaica; kurt.vaz@uwimona.edu.jm (K.V.); lennoxwaj@hotmail.com (L.A.-J.); l_latore@yahoo.com (L.L.); rory.thompson@uwimona.edu.jm (R.T.); dwight.lowe02@uwimona.edu.jm (D.L.); lowell.dilworth02@uwimona.edu.jm (L.D.); 2Department of Physical Education, Faculty of Education, The Mico University College, 1A Marescaux Road, Kingston 5, Jamaica; miller9fabian_gov@yahoo.com; 3Department of Biotechnology, Faculty of Science and Technology, The University of the West Indies, Kingston 7, Jamaica; 4School of Allied Health and Wellness, College of Health Sciences, University of Technology, Kingston 7, Jamaica; melisa_anderson4life@yahoo.com; 5Department of Basic Medical Sciences, Faculty of Medical Sciences, The University of the West Indies, Kingston 7, Jamaica; sophia.bryan04@uwimona.edu.jm; 6Caribbean Institute for Health Research, Faculty of Medical Sciences, The University of the West Indies, Kingston 7, Jamaica; shelly.mcfarlane02@uwimona.edu.jm

**Keywords:** telehealth, teleoncology, telerehabilitation, telemedicine, coronavirus disease, management, breast cancer patient

## Abstract

Telehealth is the delivery of many health care services and technologies to individuals at different geographical areas and is categorized as asynchronously or synchronously. The coronavirus disease 2019 (COVID-19) pandemic has caused major disruptions in health care delivery to breast cancer (BCa) patients and there is increasing demand for telehealth services. Globally, telehealth has become an essential means of communication between patient and health care provider. The application of telehealth to the treatment of BCa patients is evolving and increasingly research has demonstrated its feasibility and effectiveness in improving clinical, psychological and social outcomes. Two areas of telehealth that have significantly grown in the past decade and particularly since the beginning of the COVID-19 pandemic are telerehabilitation and teleoncology. These two technological systems provide opportunities at every stage of the cancer care continuum for BCa patients. We conducted a literature review that examined the use of telehealth services via its various modes of delivery among BCa patients particularly in areas of screening, diagnosis, treatment modalities, as well as satisfaction among patients and health care professionals. The advantages of telehealth models of service and delivery challenges to patients in remote areas are discussed.

## 1. Introduction

Telehealth is the provision of numerous clinical and non-clinical services, and technologies to persons at various remote locations using information and telecommunication technology (ICT) [1]. ICT is used in telehealth to deliver clinical services such as screening, diagnosis and therapy. ICT is also used for nonclinical services such as research, continuing education for health care professionals and health promotion. This results in the advancement of the health and wellbeing of persons, organizations and communities [1]. Telemedicine is a subcategory of telehealth that describes the use of telehealth technologies including electronic communication by health care providers to offer only remote clinical services to patients. In telehealth, there is no in-person visit. Technologies including secured audio and video links are used for remote clinical appointments, follow-ups, expert consultations, medication management, acute and chronic disease management, and many other clinical services [2].

The formats for telehealth comprise telephone calls for remotely connecting patients in many settings using electronic devices called peripherals (pulse oximetry devices, digital thermometer, glucose monitors or blood pressure monitors). Digital photographs and videos are also used asynchronously or synchronously in remote monitoring [3]. Synchronous telehealth occurs in real-time and uses technology such as video conferencing which permits interaction between patient and provider via a two-way audiovisual link. The health care professionals communicate directly with patient and provides medical expertise with subsequent disease diagnosis, treatment plan and medication management [4].

Asynchronous telehealth commonly called “store-and-forward telehealth” denotes a platform used in remote patient monitoring that employs technology to accumulate and transfer data (electronic health records), photographs or video information to an internet locality that can be later reviewed by the health care professional [5]. Asynchronous telehealth utilizes electronic consults which makes the patient’s electronic health record available to a specialist physician. This facilitates online interaction between the patient, referring health care provider and the specialist physician to establish a treatment plan [6]. The asynchronous telehealth electronic communications facilitate the interaction between patients and physicians as well as other health care professionals and decreases the need for referral in-person specialist appointments [6].

### Telehealth and Coronavirus Disease 2019

Prior to the coronavirus disease 2019 (COVID-19) pandemic findings from studies indicated more awareness of the use of telehealth services by health care professionals, such as physicians and nurses, as well as patients. Improvements in broadband infrastructure and internet access and connectivity have increased the delivery of telehealth. There is also enhanced availability and provision of information and communication technologies such as electronic messaging via mobile phones or computer, teleconferencing, digital monitoring, telephone or web or videoconferencing [1]. Videoconferencing is being increasingly used in the delivery of telehealth as it allows effective and reliable real-time communication between patients and health care providers. It provides visual cues which are imperative for visual assessment, clinical surveillance and sharing of clinical information pertaining to treatment plan [7].

Preceding COVID-19, a number of studies examined whether patients, physicians and caregivers were satisfied with telehealth services [8,9,10]. Martinez and colleagues performed a cross-sectional study which characterized the patterns of use of telehealth services by physicians and patients, and correlates of patient satisfaction. The majority of the 24,040 patients were mostly satisfied with direct to consumer telemedicine, and the high level of satisfaction was positively associated with coupon use and prescription receipt [10]. A systematic review of 36 studies which examined the level of satisfaction of patients and their caregivers with telehealth videoconferencing found a high level of contentment with telehealth across dimensions such as consumer focus, information sharing, system experience and overall satisfaction. The researchers indicated that patients residing in remote areas were mostly satisfied with telehealth services due to increased access to health care while decreasing possible inconvenience of travelling to a medical facility [8]. Likewise, an earlier systematic review of 24 studies showed significant improvements in engagement of caregivers of cancer patients and user satisfaction of telehealth interventions such as telephone calls and web-based platforms [9]. Another systematic review of 22 studies that examined the experiences of adult cancer survivors engaging in telehealth interventions found that these intermediations can facilitate an experience of convenience and independence, personalized care traversing physical distance and remote reassurance due to communication with health care providers [11].

The COVID-19 pandemic has resulted in significant disruptions in health care delivery and there is an increasing demand for telehealth services. Telehealth was projected to massively increase in 2020 and its use in the health care industry estimated to rise by 64.3% [12]. During 2020, telehealth rapidly grew as an exceptional integral conduit of communication between patients and health care providers and was employed in both remote and non-remote areas to enable continuity of quality care and treatment [13].

The utilization of telehealth services may diminish the possibility of transmission of severe acute respiratory syndrome coronavirus 2 (SARS-CoV-2), the virus that causes COVID-19 by limiting contact between patients and health care professionals, thereby allowing patients with the virus to remotely receive medical care [14]. Health care professionals can take on more clinical and nonclinical duties from home. Therefore, telehealth safeguards the health of both patients and health care professionals as the former may present with symptoms are screened and referred as appropriate [15]. Telehealth also delivers low-risk critical medical care to patients with non-COVID-19 conditions, and recognizes those persons who may require further clinical assessment and consultation [15].

Globally, health care systems are challenged with protecting health care workers while providing clinical services to patients suffering from COVID-19 as well as other acute and chronic disorders. Telehealth services that utilize teleconferencing and mobile apps have allowed patients with chronic conditions to be remotely monitored via virtual care with better treatment compliance and improved health outcomes [16]. There are emerging data that has documented the views of physicians with the application of telemedicine in response to the COVID-19 pandemic [17]. In a quantitative study using semi-structured interviews that examined the experience of primary care physicians and physicians-in-training with telemedicine during the COVID-19 pandemic, there was better patient access to care, additional time for counseling patients and enhanced medication reconciliation [18]. Likewise, Hasani et al. conducted a qualitative study of 22 physicians which examined their perception with the use of telemedicine in primary care using interpretive phenomenological analysis. The physicians, a majority of whom were family physicians and females, conveyed that telemedicine performed well particularly with patients suffering from COVD-19 and chronic conditions. They also indicated that there was a decrease in crowdedness in the medical facilities [19]. In another study that investigated the implementation of an app-based telemedicine program called Sofia in a primary care setting in Mexico during the COVID-19 pandemic, there was significant physician involvement, and over 80% patient satisfaction was reported subsequent to video consultation services. The authors concluded that telemedicine using video consultation and mobile applications was a safe and an effective tool to improve patients’ health during the short, medium and long terms [20].

Health care services have created innovative approaches to delivery of care to patients without compromising the health and safety of health care workers. Garg et al. conducted a prospective review of the employment of telemedicine using a synchronous audio-video communication in an outpatient adult ambulatory clinic during April 2020 and found high patient reception and satisfaction with telemedicine services. There were significant number of visits during the COVID-19 pandemic despite limitations such as shortage of providers and limited staffing [21]. A study that examined patients’ behaviors and perspective towards the use of telehealth (telephone and video component) for virtual primary care appointments at a primary and specialty care clinic, showed decreased absentee rates and significant patient satisfaction of telehealth use [22]. Furthermore, in a recent study utilizing a mixed-method approach conducted in New Zealand, it was shown that the use of telehealth services involving telephone and video consultations in primary care were associated with high patient satisfaction, convenience, safe access to health care and overall positive experience [23].

Telehealth allows safe and suitable care through decreasing physical contact at heath care facilities for cancer patients who are at risk of mortality from contracting COVID-19 [24]. A literature review examining the use of telehealth by the interdisciplinary cancer care team posited that it provides an innovative response to the challenges of offering quality and continued service to persons affected by cancer and facilitates the appropriate treatment of patients in their home environments [25]. Likewise, a literature review of 29 studies comprising of 3698 participants that examined exercise telehealth interventions using mobile applications, short messaging service (SMS), web-based platforms and telephone among cancer survivors showed reasonable compliance, relief of symptoms and a generally positive experience [26]. Narayanan et al. documented the implementation of a model of integrative oncology consultations involving non-pharmacological methods for the management of symptoms, and found that cancer survivors reported improved satisfaction and lower symptom burden compared with face-to-face prior to the COVID-19 pandemic [27].

Telehealth delivery in oncology is evolving and increasingly studies have shown its feasibility and efficacy in improving psychological and social outcomes within cancer patients [28] and access to palliative care services in a home environment [29]. An increasing number of studies have examined the effect of the COVID-19 pandemic on continued quality and appropriate care for breast cancer (BCa) survivors by qualified health care professionals in remote and non-remote settings.

This review of the literature examined the evidence of the utilization of telehealth services via its various modes of delivery among BCa patients particularly in areas of screening, diagnosis, treatment modalities and satisfaction among patients and health care workers. The benefits of telehealth models of service and delivery challenges in telehealth to patients in remote and non-remote areas are discussed and possible solutions presented.

## 2. Method

### 2.1. Study Design

A literature search was carried out by the authors to ascertain all the relevant studies on the utilization of telehealth by health care providers subsequent to COVID-19 published from 1 January 2020 to 30 June 2021. The procedural plan involved taking the following steps: (i) documentation of definite research aims and search strategies, (ii) identification and selection of peer-reviewed research articles, (iii) selection of peer-reviewed studies according to well-defined eligibility criteria and in keeping with review aims, (iv) organizing and reporting the data and finding of the peer-reviewed studies in the different sections and (v) discussion of the outcomes and conclusion.

### 2.2. Literature Search Strategies

We explored electronic databases such as PubMed, Cochrane Library, Embase, Scopus and Web of Science for possibly pertinent studies using key words and subject headings such as: telehealth, teleoncology, telerehabilitation, telemedicine, health care services, coronavirus disease, management, video conferencing, web-based platforms, breast cancer patients, screening, diagnosis, coronavirus disease 2019 (COVID-19), treatment, modalities and satisfaction.

### 2.3. Study Eligibility Criteria–Inclusion and Exclusion Criteria

The studies retrieved were carefully scrutinized to omit overlapping data or duplicates. Those studies written in other languages, no available full data and evidence in pediatric population were excluded.

The studies included in this review were published in the last 18 months in peer-reviewed journals, written in English and reported medical and scientific findings. Furthermore, relevant data that were extracted from published articles by authors comprised: first three authors, year of publication, study design, study population and information from peer-review articles concerning BCa patients with COVID-19 along with telehealth; teleoncology; surgery; telerehabilitation; mental health issues; physical exercises and satisfaction with service.

There were 111 articles identified through database including PubMed, Google Search and Cochrane Library. We excluded 26 because of duplication and 85 articles reviewed for inclusion. There were 70 full-text articles assessed for eligibility with 65 of these included in this review (Figure 1).

## 3. Results

### 3.1. Telehealth, COVID-19 and Breast Cancer Screening

The US Preventive Services Task Force endorses regular BCa screening which prior to the COVID-19 pandemic was mostly performed in a health care facility as well as in private physician offices [30]. Due to the COVID-19 pandemic and the associated risk involved, it was recommended that BCa screening with established imaging procedures should be conducted utilizing universal precautions including the wearing of personal protective equipment in order to protect health care professionals and patients from infection and minimize disease spread [31].

Globally, breast imaging units in radiology departments are challenged to maintain effective and quality service during this critical time. Salem et al. performed a retrospective study at a breast imaging unit at a university hospital from March to May 2020 and reported that there was a 73% reduction in overall breast unit activity comprising mammograms, MRIs and ultrasounds. Screenings and BCa surveillance were two of the indications for mammogram and it was noted that radiology staff made significant adjusts in ensuring continued quality care whilst maintaining patients’ safety [32]. In another study which conducted a risk assessment of the use of three BCa services including breast surgery and breast imaging in February–April 2020, there was significant decrease in breast imaging with a mean weekly reduction of 61.7% [33]. Likewise, a retrospective study that quantified imaging case volumes in a large health care system reported a decrease in breast imaging by 12.3% in 16 weeks compared with prior to COVID-19. Overall, the authors observed that in 2020, there was a reduction in mammography, ultrasound and MRI imaging by 94%, 64% and 74%, respectively [34]. Besides, there was a 40–70% reduction in radiology volume in six academic medical systems exposed to high COVID-19 surge and the most significant decrease was observed with screening mammography and smallest reduction for interventional radiology [35].

There are also other reports that have indicated the rescheduling of BCa screening leading to interruptions in disease diagnosis and treatment which could adversely affect patient outcomes in the future [36,37]. Recently, Young et al. used the OncoSim BCa microsimulation mathematical model to predict the likely long-term clinical impact of interruptions in BCa screening in Canada and reported that a 3-month interruption could result in an additional 310 cases at advanced stages and 110 cancer deaths in 2020–2029. Moreover, a 6-month interruption could result in an additional 670 cases at advanced stages and 250 cancer deaths in the same period. The authors are of the view that further decrease in case volumes and longer interruptions could lead to more cancer deaths [38]. In another study, Sharp and colleagues used the CISNET cancer simulation model to examine the likely impact of COVID-19 on BCa screening in the United States, and based on the analysis, there was a projection of about 5300 more deaths from 2020 to 2030 [39]. A later retrospective single-institution study conducted in Italy found a rise in node-positive (11.2%) and stage III BCa (10.3%) after a 2-month discontinuation of mammographic screening. The authors posited that based on the findings optimum and full capacity BCa services should be reestablished with satisfactory prioritization policies to lessen harm and satisfy the requirements for COVID-19 prevention [40].

Telehealth has become a practicable and effective tool to provide health care access to persons as it decreases face-to-face inpatient and outpatient clinical appointments for BCa screening and surveillance [25] (Figure 2). Sonagli et al. conducted a retrospective study that examined the management of outpatient medical appointments via telemedicine at a cancer center in Brazil from June to October 2020. It was found that 26.0% of the 77 patients accounted for BCa screening and 46.8% for disease follow-up. Based on the findings the authors suggested that telemedicine could be used to continue outpatient appointments for BCa screening for the period of the COVID-19 disease [41]. There is also report of the safe employment and functioning of a high quality remote teleradiology health care service within the population-based BCa program delivered by Breast Screen Australia [42]. This will allow BCa screening to continue during the COVID-19 pandemic.

The use of telemedicine in BCa screening programs have been effective in the discovery of early lesions [43]. Marino described a pilot telemedicine study involving the use of mobile units in a BCa screening program and found that 3.1% of the cases had suspicious malignant lesions, 40.8% were negative to lesions and 34.9% of cases were positive to benign lesions [44]. Chung et al. described the potential application of telemammography which could increase health care provision and access for BCa patients [45] as it facilitates communication between primary physicians and specialist, and ultimately improved patient outcomes [46]. There is also information on the implementation of a high value multisite telemammography system for real-time remote BCa patient management [47] and a novel telemammography system that utilized wavelet-based image processing procedures [48]. According to Leader et al. the implementation of a high-quality, multisite telemammography system permits the remote management of BCa patients while they remains at the health center. The procedure involved the digitization of mammography films from present and past examinations at three remote sites. The digitized mammography films along with relevant clinical information are transmitted via communication systems to the central site where they are managed and read in real time by radiology consultants [47]. The implementation of these telemammography systems during the COVID-19 pandemic should significantly improve BCa screening and diagnosis especially among patients in remote and underserved regions. This area of research is also unexplored and it is hopeful that significant work will be conducted after the COVID-19 pandemic.

### 3.2. Telehealth, COVID-19 and Breast Cancer Surgery

Globally, BCa is the most frequently diagnosed malignancy among females and the emergency of the COVID-19 pandemic has adversely affected management including therapeutic surgery. This causes delays to BCa surgery due to decreased resources at hospitals and private health care facilities including absence of surgeons, anesthesiologists and other supporting personnel, and unavailability of operating theaters and intensive care. It is well known that delayed surgery adversely affects BCa patients with increased morbidity and worse outcomes [49]. There are large retrospective population-based studies that have reported significantly reduced overall survival with greater than 30 days between BCa diagnosis and therapeutic surgery for patients with stage I and stage II disease [50,51,52] (Figure 2). There is also significantly decreased BCa-specific survival in patients with stage I BCa if the interval is beyond 60 days [50].

Telehealth is important for satisfactory continuity of care for BCa patients via therapeutic surgery and lessening the risk of exposure to the SARS-CoV-2 virus. The nature of the COVID-19 pandemic requires a change in BCa care. In response, Cadili and colleagues sought to document the effect of COVID-19 on surgery volume and changes to the care of BCa patients at a Breast Cancer Centre in Canada. The authors reported that 196 patients in 2020 compared with 129 patients in 2019 successfully underwent BCa surgery and noted significant adaptations that were implemented in response to the COVID-19 pandemic. They are of the opinion that increased access to telemedicine and better anesthetic techniques will significantly improve overall BCa surgical care during and beyond the COVID-19 pandemic [53]. Additionally, in a recent study Fregatti et al. implemented a patient-tailored screening program that sought to prevent COVID-19 infection among health care workers and BCa patients undergoing surgery. Safe BCa surgery was successfully completed for 71 patients with mean in-hospital stay of 2.2 days and no COVID-19 infection detected among staff health care workers or patients [54]. Likewise, in a retrospective observational study in Spain that examined the management of BCa patients during the COVID-19 pandemic outbreak, telemedicine was used to assess any side effects and to decrease the number of unnecessary hospital visits of patients following surgical procedures and receipt of systemic therapy [55].

The peri-operative and post-operative periods of BCa surgery may involve psychological challenges among patients where there are feelings of fear, apprehension and uncertainty. Noble et al. conducted a cross-sectional feasibility study among women prior to BCa surgery where they examined telehealth access, preferences and the level of preparedness for surgery. The majority of the participants had reasonably good internet connection and access to an appropriate electronic device for telehealth. Some of the participants expressed a need for teleconsultation particularly with providing advice regarding the surgery [56]. Moreover, in a prospective study reported by Lai et al. telehealth perioperative occupation treatment sessions particularly for 26 BCa patients living in a remote region and scheduled for surgery were conducted. The authors found that videoconferencing telemedicine for both perioperative and post-operative sessions were practical, effective and satisfactory and could be used for rehabilitative services [57] (Table 1). This practical and acceptable telehealth service utilizing a videoconferencing platform may be useful to BCa patients undergoing surgery during the COVID-19 pandemic. Notably, 79 elective BCa surgeries were successfully performed from March 15 to April 30, 2020 at the Regina Elena National Cancer Institute of Rome during a lockdown due to the COVID-19 pandemic. Follow-up was very strict and involved the use of telemedicine and messaging apps to maintain close contact with patients [58].

In April 2020 an International Collaboration Group recommended management strategies for treating cancer patients including those with breast carcinoma and suggested that telemedicine could be utilized as a valuable support to lessen the number of hospital and clinic visits and risk of exposure [59]. Colakoglu and colleagues conducted a study involving the use of video telehealth to provide consultation to patients after breast reconstruction. Eighty eight (37.4%) of the 235 BCa patients received consultation via telehealth visit relating to post-surgical wound healing. The findings of the study attest to the reliability and effectiveness of this technology to reach BCa patients in remote areas [60] (Table 1). Finally, Sharp et al. described the outpatient care of a symptomatic COVID-19 positive female after a bilateral breast reconstruction involving an autologous free tissue transfer. Telemedicine was used for draining and wound dressing patients which contributed to a positive outcome [39].

### 3.3. Teleoncology, COVID-19 and Breast Cancer

Teleoncology, an area of telehealth is the employment of telemedicine clinical oncology particularity to diagnosis (pathology and laboratory), therapy (e.g., symptom management, remote chemotherapy supervision and radiation oncology), supportive (e.g., palliative care) and prevention of cancer [61]. Teleoncology utilizes a number of technologies such as videoconferencing (e.g., Zoom and WhatsApp), clinical applications (e.g., MyChart) that enable oncologists and other specialists to provide medical care to patients in remote locations [62]. Teleoncology can use an asynchronous or synchronous format during teleconsultations and mobile applications provide support for adherence to treatment plan, lifestyle change and symptom management in a home-based care environment [62]. Teleoncology has been embraced and implemented by a number of hospitals and clinical centers to deliver cancer care to many patients and a variety of models that complement or replace face-to-face consultations and outreach services [63]. Teleoncology models have adopted web conferencing, mobile technologies and remote chemotherapy supervision models in order to improve the quality of cancer care to patients particularly in remote or rural areas [64].

Prior to the COVID-19 pandemic, the use of telehealth in clinical oncology was restricted to cancer care of patients residing in rural areas with significant approval among oncology patients and health care professionals [65]. With the COVID-19 pandemic the health care community worldwide has broadly accepted and adopted telehealth models. The implementation of telehealth has substantially decreased the risk of exposure of these cancer patients who are immunocompromised to the SARS-CO-V2 virus with less outpatient visits [66]. A study which examined the preventative measures taken to reduce the spread of SARS-CO-V-2 virus in 30 oncological care centers located in 12 countries affected by COVID-19 (using a 46-item survey) found that 90.5% implemented a triage for patients and mandatory use of personal protective equipment. Telemedicine was employed in 75.2% of the oncology centers and the authors suggested that the effectiveness of the measures should be investigated in large observational studies [67]. Moreover, a cross-sectional study that evaluated the perceptions of physicians and patients on the application of telemedicine in a radiation oncology practice found that 30.6% of patients and 67% of physicians would appreciate a more frequent use of video-consultations. In addition, the majority of patients (59.9% and 63.4%) and radiation oncologists (61.1% and 63.9%) believed that video consultations would be more useful during radiation therapy or during follow-up [68]. A recent descriptive cross-sectional study comprising cancer patients from a tertiary care comprehensive oncology unit during the COVID-19 pandemic found that the majority of patients (64.1%) who were contacted by a clinical oncology specialist using telemedicine, in particularly voice call had breast carcinoma. The authors noted that teleoncology facilitated remote communication and 32.1% of patients, including those with BCa, required no further intervention [69].

The European Society for Medical Oncology made recommendations on the management and treatment of BCa during the COVID-19 pandemic and defined three levels of priorities for clinicians namely low, intermediate and high intervention. It recommended telemedicine for patients in the low priority groups with regards to survivorship follow-up, established patients with no new issue and psychological support visits [70]. A survey conducted by the American Society for Radiation Oncology found that majority (89%) of the radiation oncology clinics offer telemedicine consultations for patients including those with BCa [71]. Additionally, virtual care is one aspect of teleoncology and the provision of virtual prescription and the subsequent delivery of medication is an alternate way to manage cancer patients including those with BCa during the COVID-19 pandemic [72]. A self-administrated web-based questionnaire survey was conducted by Tashkandi et al. that evaluated the views and awareness of medical oncologist on virtual management of cancer patients in distant geographical and rural areas, and the importance of virtually prescribing anticancer therapy during the COVID-19 pandemic. The authors reported that 82% and 75% of the oncologists were aware of virtual clinics and virtual prescription respectively, and 85% favored the virtual prescription of hormonal therapy. Of note, 50% oncologists did not favor virtual prescription of chemotherapy and 45% preferred to manage cases virtually [73] (Table 1). Additionally, Chen et al. evaluated the effectiveness and convenience of a remote oncology pharmacy service platform Cloud SYSUCC application and found that within a 6-month period, 49.2% of the remote medical prescriptions were for patients with BCa and 50.1% were hormonal drugs. The authors also indicated high satisfaction (88.0%) among the patients who utilized the remote pharmacy service platform and suggested that it is a convenient and effective means for delivering continued pharmaceutical care for patients including those with BCa during the COVID-19 pandemic [74].

There is documentation of the implementation of a virtual care program at a high-volume cancer center that was feasible, maintained care quality and outpatient caseloads and had a high satisfaction among patients and physicians [75]. Loree and colleagues carried out an international Internet-based cross-sectional survey of 381 participants between April and June 2020 that evaluated patients’ perspective and satisfaction with virtual appointment. It was noted that 21% were BCa patients actively undergoing treatment for the disease. Sixty two percent of the respondents reported having had a virtual oncology appointment, 82% had high satisfaction and there was significant use of videoconferencing [76]. Likewise, there was a recent questionnaire-based study of 215 respondents with BCa or gynecological cancer who attended an outpatient cancer clinic of which 74 had taken part in telehealth visit. The authors indicated that the majority (92%) were highly satisfied, 73% reported better access to care, 82% reported improved health and 92% saved time due to the telehealth services [77] (Table 1). Notably, telehealth interventions may afford BCa patients with remote reassurance and personalized care in the COVID-19 pandemic although each needs to be tailored to satisfy the needs of the patients [11].

### 3.4. Telerehabilitation, COVID-19 and Breast Cancer

Telerehabilitation is an evolving area of telehealth that involves providing rehabilitation services such as speech or physical therapy to patients, clients and physicians in rural and geographically remote locations using information and communication technologies [81]. Telerehabilitation provides reasonable medical care access to individuals in the comfort of their homes or other living environments, increases the reach of physicians and reduces the need for face-to-face rehabilitation particularly where there are limited personnel and financial resources [82]. Internet-based provision of telerehabilitation services may increase access and allows the rehabilitation therapist or clinician to optimize and tailor treatment in a personalized manner with resulting improved adherence and outcome [83].

BCa patients may experience significant psychological, psychosocial and physical challenges that can be appropriately addressed by rehabilitation during and after therapy which can decrease the side effects and facilitate the process of recovery [84]. There is need for BCa patients to receive adequate rehabilitation especially during the first 12 months after diagnosis and both inpatient and outpatient rehabilitative care should include motor rehabilitation, cognitive or psychological therapy and pathology-related interventions [85]. There are studies that have reported that aerobic exercise subsequent to BCa surgery can increase shoulder joint range of motion, improved pulmonary function, better quality of life (QoL) and pain relief [86,87].

The use of telehealth is a promising approach for providing support to BCa patients and there is documentation of a randomized control trial involving the use of a telerehabilitation system called e-CUIDATE to improve the QoL of survivors [88]. The same authors conducted a randomized controlled trial comprising of 81 BCa survivors who had finished adjuvant therapy for stage I to IIIA disease. Using an 8-week Internet-based, tailored exercise program, they found significantly improved scores for physical, pain severity, cognitive function, global health status, pain interference and physical role in the telerehabilitation group [88]. Furthermore, the Collaborative Care to Preserve Performance in Cancer (COPE) study which is three-arm randomized clinical trial comprising 516 patients (14% BCa) with advanced disease found that collaborative telerehabilitation moderately increased function and QoL, while reducing pain and hospital length of stay [89].

The COVID-19 pandemic has negatively impacted care for BCa patients, and Helm et al. evaluated the effect of decreased access to rehabilitative care among 15 survivors looking particularly at distress and QoL [90]. The authors reported increased distress due to the closure of rehabilitation services due to the COVID-19 pandemic was associated with decreased QoL and physical activity. There were improvements in QoL following restart of rehabilitation service and offering of telehealth in one-third of the BCa survivors [90] (Table 2).

During the COVID-19 pandemic health care facilities have been increasingly adopting telerehabilitation in preference to face-to-face physical therapy rehabilitation. De Rezende and colleagues suggested that telerehabilitation might be implemented as an initial option for continued care to BCa patients as there is decreased risk to SARS-CoV-2 [88]. The physical therapy telerehabilitation program suggested includes a range of movement modality, a multi-component exercise program and aerobic exercises that may be conducted at home [91] (Table 2). Likewise, Mella-Abarca et al. reported a model of telerehabilitation implemented in a Chilean hospital for BCa survivors. The model involves the use of telephone or video call for initial assessment and subsequent remote consultations [92] (Figure 2). Telemonitoring was conducted synchronously with the patient and information shared asynchronously via a web page. Notably, the authors indicated that there was a high level of approval and satisfaction among BCa patients, clinicians and physiotherapists [92]. The same authors described a method of telerehabilitation that involves ongoing physical therapy in a public referral hospital located in a resource-limiting setting [92] The telerehabilitation model was implemented in April 2020 and within 3 months 226 care events involving 118 BCa patients were documented of which 63% were conducted via telerehabilitation. These include pre-operative checks, follow-up appointments, prevention of lymphedema and other clinical consultations [92] (Table 2).

The information on the models of telerehabilitation and their application is an unexplored area with limited data. This warrants more large observational studies especially during the COVID-19 pandemic.

### 3.5. Telehealth, COVID-19, Breast Cancer and Mental Health Issues

A diagnosis of BCa may cause patients significant emotional, mental and psychosocial challenges which may negatively impair the ability of the patient to adequately adjust to this life changing event, resulting in reduced QoL and poor outcome [93]. There are studies that have shown that the rate of psychological dysfunction among BCa patients who experienced surgical procedures is in the range of 30–45% [94,95]. Gallagher et al. reported that 43% of BCa patients had a possible affective disorder 6 months after diagnosis and the appraisal process is supportive in the psychological adjustment to the disease [96]. There are interventions such as the online-delivered therapy called iNNOVBC, a 10 week guided Internet-delivered cognitive behavioral intervention conducted as a multi-center randomized control trial that sought to alleviate mild to moderate anxiety and depression in BCa survivors [97]. Furthermore, in an earlier randomized control trial, the intervention group exposed to an Internet-based cognitive behavioral therapy experienced significant clinical improvement with decreased severe fatigue and associated symptoms [98].

Telehealth interventions have been suggested as a pioneering and effective approach to addressing the supportive needs of BCa patients [61]. Chen et al. conducted a meta-analysis of 20 randomized controlled trials comprising 2190 BCa patients and found that telehealth intervention was concomitant with enhanced QoL and self-efficacy, lower perceived stress, depression and distress compared with usual care [99]. A recent prospective pilot study evaluated the influence of a multisite psychoeducation-based cognitive rehabilitation intervention using telehealth conferencing on 27 BCa survivors following chemotherapy. There was improvement in self-reported perceived cognitive function and high satisfaction among the participants with the secure telehealth conferencing [100].

The COVID-19 pandemic and subsequent actions are adversely affecting the QoL and daily clinical management of BCa patients. A prospective, multicenter cohort study of 1051 BCa patients and survivors that examined how COVID-19 measures affected their physical and psychological well-being as well as QoL found worsening emotional functioning and decreased social functioning for persons undergoing active treatment. There were minor improvements in physical functioning and QoL and the authors suggested that online interventions that promote good mental health and facilitate social interaction are warranted especially during periods of lockdown [101]. Likewise, a recent study conducted in the United Kingdom of 234 BCa patients revealed significantly increased COVID-19 associated anxiety, emotional vulnerability and depression among participants. The authors posited that quick application of interventions that are supportive and accessible are needed to encourage emotional resilience in the BCa populace [102].

The COVID-19 pandemic has catalyzed the application of telehealth services and other technological solutions to provide health care to patients including those with BCa. A recent study comprising 144 cancer survivors examined the impact of the COVID-19 pandemic on cancer care particularly focusing on psychological support and therapy and access to hospital-based care. The authors found that patients including those with BCa who received assistance from charity-based cancer support services following the onset of the COVID-19 pandemic had decreased QoL and non-significant increased depression, anxiety and stress [103]. Schade et al. in his editorial noted that in addition to the negative consequences of their condition, BCa patients are challenged with the psychological burden of COVID-19 and are at greater of risk of significant morbidity and mortality associated with contracting the virus. The authors suggested that telehealth, particularly teleoncology, may provide appropriate care remotely and improve mental health as it decreases stressful travel and offers a more wide-ranging approach including psychological and emotional support [104] (Figure 2).

There are very few studies that have examined the role of telehealth as a crucial tool in supporting psychiatric care of BCa patients affected by the COVID-19 pandemic. A lot of research from a mental health perspective is needed in this area.

### 3.6. Telehealth, COVID-19, Breast Cancer and Physical Exercise

Cancer patients that engage in regular exercise have better QoL and health outcomes. Notwithstanding, only 13–40% of these adult patients actively participate in physical exercise [105]. Muscle-strengthening and aerobic exercise are positively related with muscular strength and endurance, QoL as well as cardiorespiratory fitness and may afford cancer-specific survival benefits [106,107]. An International Multidisciplinary Roundtable based on evidence from the literature recommended aerobic exercise, resistance training and/or combined aerobic and resistance training for cancer patients that could increase health-related outcomes such as QoL, depressive symptoms, physical functioning, anxiety and fatigue [108]. Moreover, evidence from a systematic review of randomized control trials suggested that cancer patients who were engaged in regular physical exercise following diagnosis experienced fewer severe side effects, less morbidity and decreased relative risk cancer recurrence and mortality [109].

Over the past decade health technology including telephone, mobile and Web-based interventions have been utilized to provide support to cancer survivors [110]. A non-randomized, prospective, interventional study by Chung et al. using an Android smartphone app (WalkON^®^) was found to be effective in increasing physical exercise particularly weekly walk steps among BCa survivors in a community [111]. Additionally, telehealth interventions are increasingly being applied to remote-based exercise activities of cancer patients to promote good adherence and favorable health-related outcomes [88,112]. Larson et al. conducted a systematic and meta-analysis study that determined the benefits of telehealth-based interventions and found that compared with usual care they are effective in delivering emotional support to cancer patients including those with breast carcinoma (Figure 2). There was an overall improvement in QoL for cancer patients undergoing therapy [113]. These same authors recently conducted another systematic review and meta-analysis involving 11 articles and reported that telehealth interventions may have a positive effect on QoL of cancer survivors compared with in-person standard care [114]. Likewise, a systematic review of telephone-based and Web-based applications involving aerobic exercise, resistance training, or both demonstrated improvements in overall physical activity with most success observed with the use of phone calls intervention [115]. However, it is worth noting that a systematic review and meta-analysis by Wangwatcharakul and colleagues did not indicate any significant difference in the QoL of BCa patients receiving standard care compared with those in receipt of telemedicine [116].

The COVID-19 pandemic has negatively impacted oncology services including the delivery of in-person supervised exercised interventions to cancer patients [117]. A systematic review of 29 studies by Morrision et al. showed good compliance, symptom relief and a general positive experience using telehealth via telephone intervention, SMS messaging as well as mobile and web-based applications [26]. A recent study comprising 1315 subjects examined an integrated electronic health record-telehealth platform utilized by oncology consultants at an academic health care facility. The authors reported increased physical activity among cancer patients in the telehealth group compared with those in the in-person group [27]. Bland and colleagues are of the opinion that telehealth could be used to provide supervision of home-based exercises to BCa survivors in supportive care, especially those residing in remote locations. They noted that telehealth videoconferencing interventions that are similar in model to customary face-to-face supervised exercised interventions guided by a physical therapist would provide significant support to BCa survivors with subsequent better health-related outcomes during the COVID-19 pandemic. However, the authors mentioned limitations such as inadequate access to technology, high cost associated with specific telehealth platforms, low technology knowledge, safety concerns and absence of hands-on assistance as possible barriers to telehealth intervention design and delivery [118].

### 3.7. Implementation and Satisfaction with Telehealth/Telemedicine

Globally, the COVID-19 pandemic has negatively impacted health care systems and the diagnosis and treatment of BCa patients [119]. A survey of 404 respondents (43.5% BCa patients) showed delays or cancelled diagnostic and screening tests such as mammogram, MRI and ultrasound as well as surgery, radiation and chemotherapy visits [120]. Therefore, there is a need for subsequent restructuring of clinical activities including fundamental changes in patient management and greater application of health technology [121]. Telehealth is increasingly used in oncology practice. Hassan et al. examined the implementation of telemedicine at an Oncology Division in Israel during the COVID-19 pandemic. The study involved 172 patients, with the majority presented with gastrointestinal malignancies (46.0%), and 14.5% were BCa patients. The results suggested high patient satisfaction and perception of the technology as safe and effective (Figure 2). However, approximately one-third of the patients felt that the absence of face-to-face visits compromised their treatment although most wished to continue with telemedicine services [78]. In another recent study that examined the implementation and usage of telemedicine amidst the COVID-19 pandemic involving 1762 cancer patients, there were high levels of satisfaction among patients (92.6%) and their clinicians (65.2%) with telehealth video visits [122]. Furthermore, in a study involving 1244 BCa patients from 18 centers from Italy and France that evaluated satisfaction with telehealth visits during the COVID-19 pandemic, it was found that approval was high and the technology feasible regardless of the mode [79] (Table 1).

Telehealth is an appropriate method of providing care to cancer patients and was quickly adopted by health care facilities including oncology clinics during the COVID-19 pandemic. The COVID-19 pandemic continues to place an extraordinary demand on health care system. To maintain the prior level of quality care, the need for rapid execution of telemedicine is established. A survey of 56 cancer patients and 25 radiation oncology residents showed that majority (88%) of the participants were pleased with telehealth virtual visits and expressed that they would use this technology platform again [123]. In a study by Patt et al. that examined telemedicine in community cancer care, 76% of the 640 clinicians as well as new and established patients were satisfied with the platform and indicated that the technology allows psychological care, lessen the need for hospitalization, and provides support with clinical plan and pharmacy as well as family education [124]. Notably, in a study involving the use of a Telehealth Usability Questionnaire, telemedicine adoption and usage in clinical care of cancer patients (BCa, 36%) were perceived to be safe and favorable by physicians and patients [80] (Table 1).

Alternative health care technological models including telehealth services lessens the spread of SARS-CoV-2 virus and decreased exposure risk to the disease for clinicians and patients [66]. Likewise, an 82-item survey taken June–July 2020 among clinical oncologists in 18 countries in Europe, Latin America and United States demonstrated that surgery was the modality most significantly affected, either cancelled or delayed followed by palliative care [125]. Moreover, in a national cross-sectional community survey of 596 users in Australia, majority of the participants (62%) felt that their experience was better than traditional face-to-face medical appointments [126]. Finally, a study conducted at a multistate comprehensive cancer center of 1077 patients reported high patient satisfaction with telemedicine and significant confidence in physician and overall clinical care (Figure 2). The authors suggest that enhancement of telemedicine in oncology should be of upmost importance and improved access to audiovisual capabilities will significantly increase care by oncologists [127].

## 4. Discussion

Telehealth interventions are practical modes of health care service delivery that utilize information and communication technologies to enable patients residing in remote, rural as well as urban areas to access clinical care within their locale. These interventions apply to technological methods including electronic messaging services, videoconferencing, digital observation and mobile telephone to facilitate real-time communication between patients and health care personnel and can be tailored so that the former experiences personalized care. Two areas of telehealth that have grown significantly in the past 10 years are telerehabilitation and teleoncology [128]. Both technological systems harness the power and usability of the internet and provide opportunities at every stage of the cancer care continuum for BCa patients such as diagnosis, treatment including therapeutic exercises, surveillance as well as clinical trials in the area of clinical, radiation and surgical oncology [129,130].

Globally, COVID-19 continues to be a major health care crisis as it spreads unremittingly disrupting BCa-related care such as screening (mammography), diagnosis (diagnostic breast imaging), treatment and surveillance [131]. This has forced many health care facilities and institutions to adopt telehealth which has grown significantly from occasional to widespread use in order to deliver continuity of care in response to the COVID-19 global pandemic. With restrictions on in-person medical consultations for BCa patients, telehealth may be a worthwhile alternative and innovative approach as it provides virtual health management via videoconferencing, exercise interventions in clinical supportive care, remote pharmacy and decreased exposure to patients and health care professionals [25].

This review evaluated the literature garnered from three research databases between January 2020 and June 2021 on the implementation and application of telehealth during the current COVID-19 pandemic. Notably, the application of telehealth to oncology practice involving BCa patients have resulted in a number of benefits. Telehealth is potentially cost effective, convenient, decreases the travel burden on BCa patients for consultation with health care professional and enables the timely discussion of clinical interventions and treatment plans [65]. Telehealth facilitates real-time direct physician-patient interaction via videoconference sessions, allows greater access to clinicians in varied fields of expertise, management by a multidisciplinary clinical team and easier transmission of clinical information including imaging, laboratory and histopathological data. The application of telehealth also comprises distant monitoring of the side effects of treatment, symptom management, emotional and psychological support, home-based individually tailored exercise program under the guidance of a professional, enrollment and participation in clinical trial [132].

Although there are many benefits to BCa patients from the application of telehealth, there are many challenges and concerns among patients, and health care workers and providers of service. Prior to the COVID-19 pandemic, telehealth visits via videoconferencing were less common due to reluctance to use the technology particularly by older patients due to preference for face-to-face consultations, unstable internet connections in remote areas, patient distrust as it is not possible to perform remotely a full physical examination, and lack of insurance reimbursement [133]. There are other challenges with telehealth including concerns regarding security of patient health records transmitted electronically, high cost associated with acquisition and implementation, significant maintenance costs, administration and training of health care professionals to effective utilize the different platforms, inadequate access to technology or little platform literacy and the inattentiveness of physicians during clinical consultations [134].

There are possible solutions to the challenges of implementing and providing an efficient telehealth services particularly among cancer patients who are older, with low income, underinsured and overall low socio-economic status as well as geographic limitations. Access could be improved by the implementation of clinical video telehealth clinics by private and public healthcare providers in rural and remote areas with increasing use of asynchronous and synchronous photographic and video formats. A good example is a report in 2018 where the Pittsburgh Healthcare System of the United States Department of Veterans Affairs (VA) established remote video telehealth clinics within the Virtual Cancer Care Network which allowed the virtually delivery of care to cancer patients by oncologists at the tertiary center [135]. In overcoming barriers to access, VA partnered with T-Mobile and Phillips Healthcare to provide patients with VA telehealth services. For patient in rural or underserved locations, VA partnered with Walmart to deliver telehealth services via technological systems placed in retail stores [136].

With the widespread implementation of telehealth due to the global COVID-19 pandemic, there is relaxation of restraining regulations for telehealth deployment by governments and increasing coverage of the cost of telehealth visits by insurance companies [77]. In improving reimbursement for clinician–patient video visits there need to be negotiations between insurance companies and private healthcare systems regarding suitable payment models which is aligned with the telehealth service offered. It is important that reimbursement to physicians for patient care provided via telehealth is at the same rate as face-to-face visits.

There should be the requisite infrastructure for wireless connectivity at both ends of the patient-physician encounter. In the clinical encounter via telehealth, the continuity of care involves the use of electronic health records. Given the challenges with privacy and security of patient information, telehealth services should satisfy the requirements of the Health Insurance Portability and Accountability Act (HIPAA). This will ensure that patient information is confidential and secured in addition to the other cybersecurity measures implemented when the telehealth services were established. In lieu of the online security concerns, medical centers should use webinar platforms that are HIPAA compliant.

Nevertheless, despite these challenges, telehealth continues to deliver timely and appropriate solutions to the obstacles caused by the COVID-19 pandemic on the delivery of interdisciplinary cancer services for BCa patients by a multidisciplinary health care team.

## 5. Conclusions

Telehealth may be a practical technological approach to the continuity of medical care for BCa patients during the current COVID-19 pandemic. The evidence from the studies demonstrates the use of telemedicine, teleoncology and telerehabilitation, three divisions of telehealth on various platforms in the continued care of BCa patients with resulting improved patient outcomes. Though there is reasonably good patient acceptance and satisfaction, there are a number of barriers to overcome and the need to conduct more observational studies with larger sample size among participants in remote areas with reliable internet access.

## 6. Future Perspective

In order to improve the patient-physician interaction challenges associated with virtual management via videoconferencing should be addressed to advance cancer care and better patient outcomes. There are ongoing revisions and an upgrading of telehealth protocols which should be tested using more rigorous randomized designs with adequate number of participants.

While there is increasing research in the areas of telemedicine, teleoncology, telerehabilitation, further worldwide research is warranted to implement telehealth in primary care and psychiatry, and monitor specific quality and efficiency indicators. Moreover, prospective telehealth interventions should be established iteratively in partnership with an extensive range of cancer survivors including BCa patients to maximize patient–physician engagement and overall patient benefit. The continuing implementation and improved use of telehealth services requires collaborative research on the application of technological and clinical specialty best practices and addressing privacy concerns so that optimum benefits are derived by the patients and other users such as the health care providers.

Likewise, there also needs to be more research in the application of telehealth concerning the emotions and perspective of cancer patients including those with breast carcinoma, and patient–clinician verbal and non-verbal communication in everyday oncology practice to minimize distrust and emotional disconnection particularly among older patients.

Finally, given the current COVID-19 pandemic, more investigation is required to evaluate the viability and patient experience of platforms of telehealth including FaceTime, Skype, Microsoft Teams and Zoom. There should be current guidelines for the ethical use of telehealth support for physicians and health care organizations and the findings from the research should be used by policy-makers and decision-makers to inform the way forward.

## Figures and Tables

**Figure 1 healthcare-09-01401-f001:**
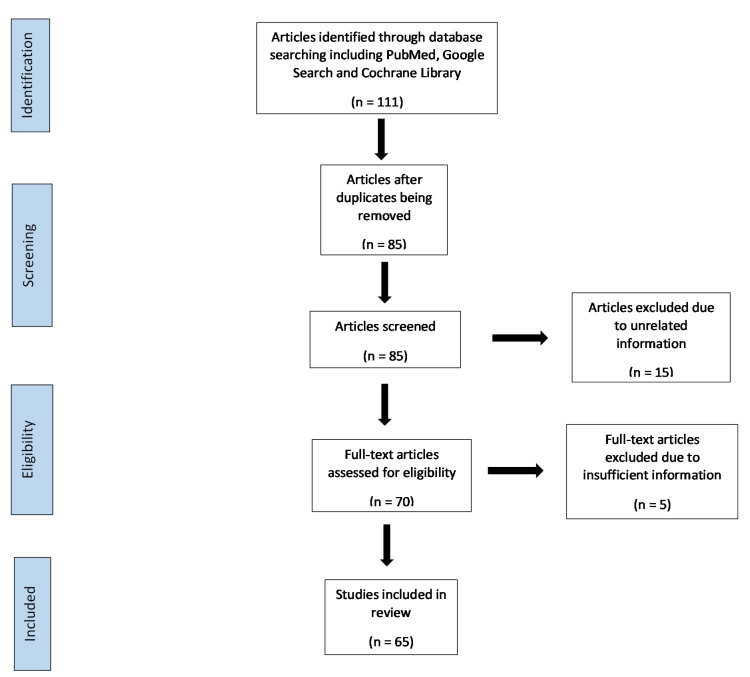
Flow diagram showing the article selection process.

**Figure 2 healthcare-09-01401-f002:**
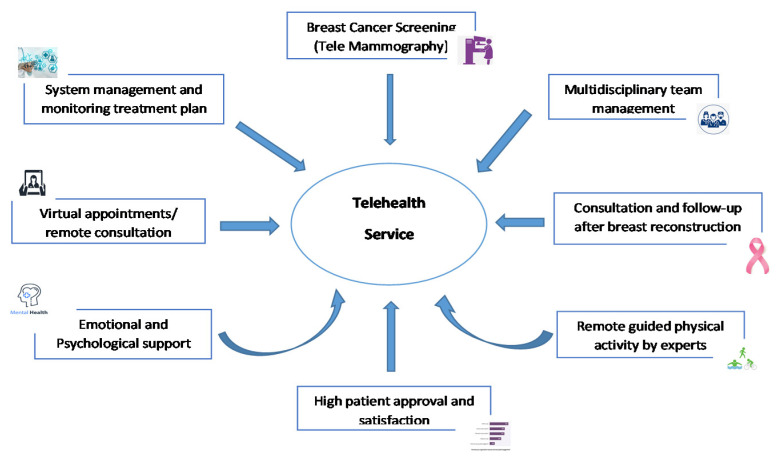
Schematic diagram of the utilization of telehealth to manage breast cancer patients.

**Table 1 healthcare-09-01401-t001:** The utilization and benefits of telehealth and telemedicine to breast cancer patients during the COVID-19 pandemic.

Health Care Technological Tool	Study Design–Number of Participants	Objective of the Study	Main Findings	References
Telehealth	Retrospective study of 1351 subjects (842 in-person and 509 telehealth)	To assess the feasibility of conducting integrative oncology consultations via telehealth during the COVID-19 pandemic (2020)	Increased physical activity among cancer patients in the telehealth group compared with those in the in-person group.	Narayanan et al., 2020 [27]
Telemedicine	Retrospective study of 77 cancer patients at Oncology Center in Brazil	To observe the use of telemedicine appointments at Oncology Centre	Telemedicine appointments for breast cancer follow-up (46.8%), breast cancer screening (26%) and benign breast disease evaluation (13%).	Sonagli et al., 2021 [41]
Telemedicine	Prospective study of 26 female breast cancer patients	To evaluate the viability and acceptance of occupational therapy services using a telemedicine model	Videoconferencing telemedicine for both perioperative and post-operative sessions were practical, effective and satisfactory and could be used for rehabilitative services.	Lai et al., 2021 [57]
Telehealth	Prospective study of 235 breast reconstruction surgery patients	To investigate the use of video telehealth to provide consultation to patients after breast reconstruction	Eighty eight (37.4%) of the 235 breast cancer patients received consultation via telehealth visit relating to post-surgical wound healing.	Colakoglu et al., 2021 [60]
Telehealth	A self-administrated electronic survey of 222 medical oncologist including those treating breast cancer patients	To evaluate the views of oncologists on virtual management of patients and the priority of prescribing anti-cancer treatments	Clinical oncologists have a high level of awareness of virtual management but 50% did not favor virtual prescription of chemotherapy and 45% prefer to manage cases virtually.	Tashkandi et al., 2020 [73]
Telehealth	Questionnaire-based study of 215 respondents with breast or gynecological cancer; 74 participated in telehealth visit	To assess the perceptions of the utility of telehealth among cancer patients in an outpatient breast/ gynecological centre	Majority (92%) was highly satisfied, 73% reported better access to care, 82% improved health and 92% saved time due to the telehealth services.	Zimmerman et al., 2020 [77]
Telehealth	Study involved 172 patients with the majority presenting with gastrointestinal malignancies (9.5%) with 14.5% breast cancer patients	To evaluate patients’ perspectives and preferences regarding telemedicine	High patient satisfaction and perception of the technology as safe and effective.	Hasson et al., 2021 [78]
Telehealth	A study involving 1244 breast cancer patients from 18 centers from Italy and France	To assess the levels of satisfaction of clinicians and patients who agreed to video visits	High satisfaction with telehealth visits during the COVID-19 pandemic, and the technology was feasible regardless of the mode.	Bizot et al., 2020 [79]
Telemedicine	Survey of 105 patients (38%)	To assess usability of virtual cancer care delivery for patients and providers across	Telemedicine adoption and use in clinical care of cancer patients was perceived to be safe and favorable by physicians and patients.	Miller et al., 2020 [80]

**Table 2 healthcare-09-01401-t002:** The utilization and benefits of teleoncology and telerehabilitation to breast cancer patients during the COVID-19 pandemic.

Healthcare Technological Tool	Study Design–Number of Participants	Objective of the Study	Main Findings	References
Teleoncology	Descriptive cross-sectional study of 421 cancer patients	The use of teleoncology (video call) at a tertiary care comprehensive oncology center	Majority of patients (64.1%) who were contacted using telemedicine had breast cancer. Teleoncology facilitated remote communication and 32.1% of patients required no further intervention.	Yildiz and Oksuzoglu, 2020 [69]
Teleoncology	Internet-based cross-sectional survey of 381 participants (21% breast cancer patients)	To evaluate the perspective and satisfaction of cancer patients with virtual appointment	Sixty two percent of the respondents reported having had a virtual oncology appointment, 82% had high satisfaction and there was significant use of videoconferencing.	Loree et al., 2021 [76]
Telerehabilitation	Survey of 15 women with primary complaints of shoulder stiffness, pain, and lymphedema	To evaluate distress and quality of life of breast cancer patients receiving care for breast cancer–related impairments during closure of rehabilitation services due to COVID-19.	Increased distress due to the closure of rehabilitation services due to the COVID-19 pandemic which was associated with decreased quality of life and physical activity. There were improvements following restart of rehabilitation service and offering of telehealth to one-third of the breast cancer survivors.	Helm et al., 2020 [90]
Telerehabilitation	Study of breast cancer survivors that require rehabilitation as part of their care	To implement telerehabilitation as an initial option for continued care to breast cancer patients	The physical therapy telerehabilitation program provides a range of movement modality, a multicomponent exercise program and aerobic exercises that could be performed in a home environment.	de Rezende et al., 2021 [91]
Telerehabilitation	Study of breast cancer survivors that attend a Chilean hospital	To implement a model of telerehabilitation	High level of approval and satisfaction by breast cancer patients, clinicians and physiotherapists.	Mella-Abarca et al., 2020 [92]
Telerehabilitation	Study conducted over approximately 3 months comprising 118 breast cancer patients	To implement a model of telerehabilitation in a public referral hospital	Majority (63%) of the 226 events were conducted via telerehabilitation. These include pre-operative checks, follow-up appointments and prevention of lymphedema.	Mella-Abarca et al., 2020 [92]

## Data Availability

Not applicable.

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
