# Peer review of "The Utilization and Benefits of Telehealth Services by Health Care Professionals Managing Breast Cancer Patients during the COVID-19 Pandemic"

_healthcare, 2021, doi:10.3390/healthcare9101401_

Round 1

Reviewer 1 Report

Comments on “The Utilization and Benefits of Telehealth Services by Health Care Professionals Managing Breast Cancer Patients During the COVID-19 Pandemic”

The Coronavirus Pandemic (COVID-19) has inevitably affected the conditions of healthcare provision. Many health services have shifted to telecommunications system to minimize exposures to infectious agents. The domain regarding telehealth is no longer preliminary, as numerous studies have investigated the effect of such a technology's presence from both patients and healthcare providers’ perspectives. Below are suggestions to further strengthen the manuscript.

  1. Conclusion indicates: While there is increasing research in the areas of telemedicine, teleoncology, telerehabilitation, further worldwide research is warranted to implement telehealth in primary care and psychiatry, and monitor specific quality and efficiency indicators.( Line 708, 709). However, the authors have surveyed the efficiency indicators as Lines 535 and 536 present. Cancer patients that engage in regular exercise have better QoL and health outcomes. (Line535) Notwithstanding, only 13-40% of these adult patients actively participate in physical exercise (Line 536) [102]. Publications on Quality of life indicator from reality on primary care through the telecommunications system do have existed before and during the pandemic.
  2. Based on the review of existing publications, readers are expected to see the suggestions or the answers on the existing difficulties, which are not mentioned in this manuscript. The contributions could be limited.
  3. The grammer, punctuation, and word sizes are not consistent in the content, such as P1. Line 21, P. 2 Line 71-74, P11. line 107-108, P. 16 line 648, and so on.
  4. Authors are suggested to go through the whole manuscript before submission.

Author Response

Reviewer 1

Comments and Suggestions for Authors

Comments on “The Utilization and Benefits of Telehealth Services by Health Care Professionals Managing Breast Cancer Patients During the COVID-19 Pandemic”

The Coronavirus Pandemic (COVID-19) has inevitably affected the conditions of healthcare provision. Many health services have shifted to telecommunications system to minimize exposures to infectious agents. The domain regarding telehealth is no longer preliminary, as numerous studies have investigated the effect of such a technology's presence from both patients and healthcare providers’ perspectives. Below are suggestions to further strengthen the manuscript.

  1. Conclusion indicates: While there is increasing research in the areas of telemedicine, teleoncology, telerehabilitation, further worldwide research is warranted to implement telehealth in primary care and psychiatry, and monitor specific quality and efficiency indicators.( Line 708, 709). However, the authors have surveyed the efficiency indicators as Lines 535 and 536 present. Cancer patients that engage in regular exercise have better QoL and health outcomes. (Line535) Notwithstanding, only 13-40% of these adult patients actively participate in physical exercise (Line 536) [102]. Publications on Quality of life indicator from reality on primary care through the telecommunications system do have existed before and during the pandemic.

Response

The authors took note of the statement by the reviewer, “Publications on Quality of life indicator from reality on primary care through the telecommunications system do have existed before and during the pandemic”.

  1. Based on the review of existing publications, readers are expected to see the suggestions or the answers on the existing difficulties,which are not mentioned in this manuscript. The contributions could be limited.

Response

  • The challenges of telehealth (i.e. privacy, information security, access, reimbursement, etc) are given on Page 15 of 25 (highlighted in green).
  • Solutions to these challenges are provided on Page 16 of 25 (highlighted in blue).
  1. The grammer, punctuation, and word sizes are not consistent in the content, such as P1. Line 21, P. 2 Line 71-74, P11. line 107-108, P. 16 line 648, and so on.

Response

  • Line 21. The word size was adjusted and is the same as other parts of the manuscript.
  • Line 71 – 74. The word size was adjusted and is the same as other parts of the manuscript.
  • Line 107-108. The word size was adjusted and is the same as other parts of the manuscript.
  • Line 648. The corrections regarding spacing and word size were done.
  • The entire manuscript was reviewed and corrected for grammar, punctuation and word size.
  1. Authors are suggested to go through the whole manuscript before submission.

Response

  • The authors carefully and thoroughly reviewed the manuscript for grammatical errors etc. Changes/corrections are in red and highlighted in yellow.
  • Additional information requested by other reviewers (such as in the discussion is highlighted in blue).

Reviewer 2 Report

In the manuscript, the authors indicate that they have conducted a systematic review of literature on telehealth services among breast cancer (BCa) patients, particularly in the areas of screening, diagnosis, treatment modalities, and patient and provider satisfaction.  The articles selected and described in the manuscript were published during the coronavirus pandemic from January, 2020 to June, 2021, with the apparent attempt to describe telehealth services for breast cancer patients during the pandemic when there were major disruptions in health care delivery for breast cancer patients. 

This manuscript describes results from a wide range of articles.  Given the various outcomes that are described, it probably should not be called a “systematic review.”  The eligibility criteria for selecting articles does not really fit the rigorous process used for systematic reviews.  It is simply a literature review of telehealth services for breast cancer patients that occurred during the pandemic, with inconsistent attention paid to “outcomes” in the studies. Outcomes ranged from patient and provider satisfaction to percentages of patients who were evaluated or received a consultation via telemedicine. 

The descriptive material of results from the articles is organized into sections, which is helpful.  The reader does come away with a feeling for how providers used telehealth services to diagnose, offer consultations, order prescriptions, manage treatment, monitor patients post-surgery, and offer telerehabilitation services.

Suggestions for further strengthening the manuscript follow:

  1. The tables could use more clarity so that the titles clearly indicate what is displayed in the tables. It might help to provide a table for each section that has its own heading.
  2. It is unclear why only a subset of the 65 articles selected for this review are displayed in the two tables.
  3. There needs to be more specific information to help the reader understand what is meant by tele-mammography. Presumably mammography involves an in-person procedure.  It’s hard to envision how tele-mammography could be used in lieu of in-person mammography. 
  4. The writing needs work. Many sentences are long or awkward, verb usage (plural vs. singular) is not correct, etc.  Below are some examples, but someone with expertise in editing in English needs to carefully edit the entire manuscript.

- line 611.  The sentence with the word “repeated” is very awkward.  It should probably read “expressed that they would use this technology platform again”

- line 605.  “proving” should read “providing”

-  Line 569-570.   “relied” should read “relief”

Author Response

Reviewer 2

Comments and Suggestions for Authors

In the manuscript, the authors indicate that they have conducted a systematic review of literature on telehealth services among breast cancer (BCa) patients, particularly in the areas of screening, diagnosis, treatment modalities, and patient and provider satisfaction.  The articles selected and described in the manuscript were published during the coronavirus pandemic from January, 2020 to June, 2021, with the apparent attempt to describe telehealth services for breast cancer patients during the pandemic when there were major disruptions in health care delivery for breast cancer patients. 

This manuscript describes results from a wide range of articles.  Given the various outcomes that are described, it probably should not be called a “systematic review.”  The eligibility criteria for selecting articles does not really fit the rigorous process used for systematic reviews.  It is simply a literature review of telehealth services for breast cancer patients that occurred during the pandemic, with inconsistent attention paid to “outcomes” in the studies. Outcomes ranged from patient and provider satisfaction to percentages of patients who were evaluated or received a consultation via telemedicine. 

The descriptive material of results from the articles is organized into sections, which is helpful.  The reader does come away with a feeling for how providers used telehealth services to diagnose, offer consultations, order prescriptions, manage treatment, monitor patients post-surgery, and offer telerehabilitation services.

Suggestions for further strengthening the manuscript follow:

  1. The tables could use more clarity so that the titles clearly indicate what is displayed in the tables. It might help to provide a table for each section that has its own heading.

Response

  • The correction was made regarding the manuscript being a literature view (Page 4 of 25).
  • The authors did some modification to the tables. Table 1 contains information on studies involving telehealth and telemedicine, while table 2 teleoncology and telerehabilitation. There is not enough studies with significant findings to enable separate tables for teleoncology and telerehabilitation.
  1. It is unclear why only a subset of the 65 articles selected for this review are displayed in the two tables.

 Response

  • The authors carefully examined the 65 articles from primary studies and selected the ones that presented significant findings due to the utilization of telehealth, telemedicine, teleoncology and telerehabilitation.
  1. There needs to be more specific information to help the reader understand what is meant by tele-mammography. Presumably mammography involves an in-person procedure.  It’s hard to envision how tele-mammography could be used in lieu of in-person mammography. 

Response

  • The following information on telemammography was added to page 7.

According to Leader et al. the implementation of a high-quality, multisite telemammography system permits the remote management of BCa patients while they remains at the health center.  The procedure involved the digitization of mammography films from present and past examinations at three remote sites. The digitized mammography films along with relevant clinical information are transmitted via communication systems to the central site where they are managed and read in real time by radiology consultants [47].

  1. The writing needs work. Many sentences are long or awkward, verb usage (plural vs. singular) is not correct, etc.  Below are some examples, but someone with expertise in editing in English needs to carefully edit the entire manuscript.

- line 611.  The sentence with the word “repeated” is very awkward.  It should probably read “expressed that they would use this technology platform again”

- line 605.  “proving” should read “providing”

-  Line 569-570.   “relied” should read “relief”

Response

  • The authors carefully and thoroughly reviewed the manuscript for grammatical errors etc. Changes/corrections are in red and highlighted in yellow.
  • Line 611 was corrected as suggested by reviewer.
  • Line 607 was corrected as suggested by reviewer.
  • Line 569 – 570. The correction was made as suggested by reviewer.
  • The authors carefully and thoroughly reviewed the manuscript for grammatical errors etc. Changes/corrections are in red and highlighted in yellow.

Reviewer 3 Report

This manuscript provides a review of the literature related to benefits of telehealth services for breast cancer (BCa) care and management and impact  and opportunities of telehealth during the COVID-19 pandemic. Overall, the review is well done with a comprehensive discussion of various aspects of how telehealth can support BCa management (cancer screening/diagnosis, surgery, therapy, support, rehabilitation, exercise). As a review paper, there is limited novelty to the manuscript and perhaps a criticism may be that it descriptive in nature and lacks concrete actionable suggestions.   Regardless, the contents are still relevant and useful for healthcare practice.  Some minor suggestions/criticisms include:

1) Introduction: Pg 2, Ln 46: The terms telehealth and telemedicine are often used interchangeably. The definition given for "telemedicine" seems redundant with that given for "telehealth" (Pg1, Ln 40) rather than as a "sub-category of telehealth". Perhaps some additional clarification is needed. As a sub-category, the term telemedicine more closely aligns with the asynchronous telehealth description given in Pg 2, ln 60-63). 

2) Method. This reviewer suggests including "Study eligibility criteria - Inclusion and exclusion criteria" and Fig 1 under the Methods section (2.3 ?) as it is a part of the methodology for literature selection.

3) Results: Suggest sections 3.1-3.7 might fall under a heading of Results  (Section 3.0) something similar. 

   * For 3.1 (screening), 3.2 (surgery), 3.3 (teleoncology), it is unclear how telehealth addresses the gaps caused by COVID-19. For example, telehealth cannot facilitate at-home screening or surgery to make up for the dramatic reduction in mammographies (-94%), ultrasound (-64%) and MRI (-74%) performed (ref 34). Similarly, how does telehealth overcome the reduction in surgeries performed as a result of COVID-19 and facility access issues?  Similarly, the teleoncology discussion mainly relates to satisfaction (pg 8, ln 366-377) with telehealth rather than efficacy and clinical care. Justification for a telehealth approach in these sections is not strong.

3) On pg 6, there is a reference to Figure 2 which doesn't show up until pg 16. It is unclear if there was a different Fig 2 in another version.  Please check this reference. If correct, it would be appropriate to move Fig 2 closer to where it is first referenced.

4) Tables 1 and 2 seem to be redundant with the text describing it and do not add much to the manuscript. 

5) In several places in 3.4-3.6, it seems there is a lack of objective data to support the conclusions put forth. Perhaps, this is because the current review includes other literature reviews of 1st sources. In any case, it would be good to review the original research papers rather than reciting reviews done by others.

6) The discussion does not seem to present any new knowledge in the field of telehealth and seems to repeat many of the prior literature reviews in the benefits of telehealth (increased access, potential benefit in cost-effectiveness... although no evidence is presented, improved convenience, decreased travel, etc).  Perhaps inclusion of the drawbacks  and challenges of telehealth is warranted (i.e. privacy, information security, access, reimbursement, etc) in order to give a more balanced review.  Much of this latter discussion is missing and remains unresolved.

Author Response

Reviewer 3

Comments and Suggestions for Authors

This manuscript provides a review of the literature related to benefits of telehealth services for breast cancer (BCa) care and management and impact  and opportunities of telehealth during the COVID-19 pandemic. Overall, the review is well done with a comprehensive discussion of various aspects of how telehealth can support BCa management (cancer screening/diagnosis, surgery, therapy, support, rehabilitation, exercise). As a review paper, there is limited novelty to the manuscript and perhaps a criticism may be that it descriptive in nature and lacks concrete actionable suggestions.   Regardless, the contents are still relevant and useful for healthcare practice.  Some minor suggestions/criticisms include:

1) Introduction: Pg 2, Ln 46: The terms telehealth and telemedicine are often used interchangeably. The definition given for "telemedicine" seems redundant with that given for "telehealth" (Pg1, Ln 40) rather than as a "sub-category of telehealth". Perhaps some additional clarification is needed. As a sub-category, the term telemedicine more closely aligns with the asynchronous telehealth description given in Pg 2, ln 60-63). 

Response

The points made by the reviewer is noted.

The authors recognized that telemedicine is a sub-category of telehealth.  Telehealth is different from telemedicine in that it refers to a broader scope of remote health care services than telemedicine. Telemedicine refers specifically to remote clinical services, while telehealth can refer to remote non-clinical services.

The changes made in clarifying telehealth and telemedicine in the sentences are highlighted in blue.

2) Method. This reviewer suggests including "Study eligibility criteria - Inclusion and exclusion criteria" and Fig 1 under the Methods section (2.3 ?) as it is a part of the methodology for literature selection.

Response

"Study eligibility criteria - Inclusion and exclusion criteria" and Fig 1 were included under the Methods section as it is a part of the methodology for literature selection (as recommended by reviewer).

3) Results: Suggest sections 3.1-3.7 might fall under a heading of Results  (Section 3.0) something similar. 

   * For 3.1 (screening), 3.2 (surgery), 3.3 (teleoncology), it is unclear how telehealth addresses the gaps caused by COVID-19. For example, telehealth cannot facilitate at-home screening or surgery to make up for the dramatic reduction in mammographies (-94%), ultrasound (-64%) and MRI (-74%) performed (ref 34). Similarly, how does telehealth overcome the reduction in surgeries performed as a result of COVID-19 and facility access issues?  Similarly, the teleoncology discussion mainly relates to satisfaction (pg 8, ln 366-377) with telehealth rather than efficacy and clinical care. Justification for a telehealth approach in these sections is not strong.

Response

  • The authors creates 3.0 Results and all the sections 3.1-3.7 now falls under the heading of Results.
  • The other comments by the reviewer are noted.

3) On pg 6, there is a reference to Figure 2 which doesn't show up until pg 16. It is unclear if there was a different Fig 2 in another version.  Please check this reference. If correct, it would be appropriate to move Fig 2 closer to where it is first referenced.

Response

  • Figure 2 that was first mentioned on Page 6 was moved to page 7.
  • Figure 2 is mentioned on pages 6, 7, 11, 12, 13 and 4.

4) Tables 1 and 2 seem to be redundant with the text describing it and do not add much to the manuscript. 

Response

- The authors are of the opinion that Table 1 and 2 does add value to the manuscript.  It provides information on key studies to the readers on telehealth, telemedicine, teleoncology and telerehabilitation.  The authors did some modification to the tables based on a recommendation from one of the reviewers.  Table 1 contains information on studies involving telehealth and telemedicine, while table 2 teleoncology and tele rehabilitation.

5) In several places in 3.4-3.6, it seems there is a lack of objective data to support the conclusions put forth. Perhaps, this is because the current review includes other literature reviews of 1st sources. In any case, it would be good to review the original research papers rather than reciting reviews done by others.

Response

  • The information in section 3.4 - 3.6 were taken from primary sources rather than from secondary ones.

6) The discussion does not seem to present any new knowledge in the field of telehealth and seems to repeat many of the prior literature reviews in the benefits of telehealth (increased access, potential benefit in cost-effectiveness... although no evidence is presented, improved convenience, decreased travel, etc).  Perhaps inclusion of the drawbacks  and challenges of telehealth is warranted (i.e. privacy, information security, access, reimbursement, etc) in order to give a more balanced review.  Much of this latter discussion is missing and remains unresolved.

Response

  • The challenges of telehealth (i.e. privacy, information security, access, reimbursement, etc) are given on Page 15 of 25 (highlighted in green).
  • Solutions to these challenges are provided on Page 16 of 25 (highlighted in blue).

Round 2

Reviewer 1 Report

No other comments.